fMRI-based Alzheimer’s disease detection via functional connectivity analysis: a systematic review

Alarjani Maitha 221400375@student.kfu.edu.sa
Almarri Badar
Department of Computer Science, King Faisal University , Alhsa , Saudi Arabia
Balas Valentina Emilia
Electronic publication date: 2024 Oct 16
Publication date: 2024
Volume: 10
Electronic Location ID: e2302
Received 2024 Feb 2; Accepted 2024 Aug 12
Copyright: ©2024 Alarjani et al.
Copyright year: 2024
Copyright holder: Alarjani et al.
License: This is an open access article distributed under the terms of the Creative Commons Attribution License, which permits unrestricted use, distribution, reproduction and adaptation in any medium and for any purpose provided that it is properly attributed. For attribution, the original author(s), title, publication source (PeerJ Computer Science) and either DOI or URL of the article must be cited.
License URL: https://creativecommons.org/licenses/by/4.0/

Keywords: Alzheimer’s disease, Functional connectivity, Neuroimaging, Resting-state functional magnetic resonance imaging, Brain networks, Default mode network, Feature extraction, Brain regain

Funding: The Deanship of Scientific Research, Vice Presidency for Graduate Studies and Scientific Research, King Faisal University, Saudi Arabia Grant No. 4362 This work was supported by the Deanship of Scientific Research, Vice Presidency for Graduate Studies and Scientific Research, King Faisal University, Saudi Arabia (Grant No. 4362). The funders had no role in study design, data collection and analysis, decision to publish, or preparation of the manuscript.

==============================
Alzheimer’s disease is a common brain disorder affecting many people worldwide. It is the primary cause of dementia and memory loss. The early diagnosis of Alzheimer’s disease is essential to provide timely care to AD patients and prevent the development of symptoms of this disease. Various non-invasive techniques can be utilized to diagnose Alzheimer’s in its early stages. These techniques include functional magnetic resonance imaging, electroencephalography, positron emission tomography, and diffusion tensor imaging. They are mainly used to explore functional and structural connectivity of human brains. Functional connectivity is essential for understanding the co-activation of certain brain regions co-activation. This systematic review scrutinizes various works of Alzheimer’s disease detection by analyzing the learning from functional connectivity of fMRI datasets that were published between 2018 and 2024. This work investigates the whole learning pipeline including data analysis, standard preprocessing phases of fMRI, feature computation, extraction and selection, and the various machine learning and deep learning algorithms that are used to predict the occurrence of Alzheimer’s disease. Ultimately, the paper analyzed results on AD and highlighted future research directions in medical imaging. There is a need for an efficient and accurate way to detect AD to overcome the problems faced by patients in the early stages.

Introduction

In Alzheimer’s disease (AD), an individual’s capability to perform daily activities is affected due to the neurodegenerative disorder. This is due to the functionality problem of cognition. This is a progressive and irreversible brain disorder that slowly destroys thinking capabilities, resulting in a loss of cognitive function. It also damages the memory of the affected person by gradually damaging the neurons. AD is one of the illnesses that cause brain damage (Goedert & Spillantini, 2006). It has a long history of disturbing the quality of life of older people, and in recent decades, it has become a leading cause of death (Lane, Hardy & Schott, 2018). AD was discovered for the first time by the German disease scientist and psychological doctor Alieis Alzheimer in 1906 and spread worldwide (Cipriani et al., 2011). This disease has three stages: normal controls (NC), mild cognitive impairment (MCI), and AD. MCI is the intermediate stage of this disease. There is no cure for AD, and the quality of patients’ lives can be improved by diagnosing it early (Alzheimer’s Association, 2015; Saraiva et al., 2016). Much importance has been given to this disease by neurologists, neuroradiologists, and neuroscientists due to the aging of people around the globe.

According to the Richens, Lee & Johri (2020), 5% of outpatients are wrongly diagnosed, which affects critical cases. Inaccurate or misleading results may cause the risk of a patient’s life. According to another research, around 20% of primary care patients were also wrongly diagnosed and 33% of those patients were in critical condition. As this is an uncurable disease, each year numerous people die due to this disease. According to statistics from the World Health Organization (WHO), more than 35.6 million people have been diagnosed with dementia since 2012, with Alzheimer’s accounting for more than 70% cases. This number is predicted to triple by 2050 World Health Organization (2022). This disease is also referred to as behavioral disturbance based on intellectual and behavioral deficits. There is a need for an accurate system that can be effectively used in early-stage diagnosis.

The diagnosis can be expressed as various certainty degrees as possible. There is also an unambiguous diagnosis technique for AD with the help of an autopsy of brain tissues by demonstrating pathological changes (McKhann et al., 1984; McKhann et al., 2011). The earliest manifestation and clinical hallmark of AD is memory impairment based on repetitive episodes. Several more deficits are also presented in the patient’s language, orientation, executive functions, productive capabilities, and perceptual capabilities. The symptoms based on behavior or psychological changes are based on irritability, apathy, depression, delusions, anxiety, disinhibition, hallucinations, and eating and sleeping changes. These symptoms lead to impairments in social and professional activities (Association, 2000).

The diagnosis techniques can be classified into two types such as invasive and non-invasive. In invasive techniques, the biomarkers are used to diagnose AD, while many non-invasive methods are being utilized to detect AD, such as speech/voice and language. For early indicators of AD, a person’s cognitive performance, language difficulty, and mainly forgetting the names of familiar people or objects are key indicators (Vigo, Coelho & Reis, 2022). Also, AD can be detected via the retina as the retina reflects the changes in the brain. Both are obtained from the exact embryological origin; researchers extend from the neural tube (den Haan et al., 2017). While this is considered a new and ineffective technique for diagnosis, moreover, in most cases, AD is caused by one or more genetic factors (e.g., amyloid). This can either increase or decrease the chance of disease (Agarwal & Khan, 2020). Recent diagnosis techniques are based on the detection of AD through neuroimaging. Generally, it refers to any brain scanning technique that displays the structure and functionality of the brain (Weiner et al., 2010).

The underlying causes of AD are still unclear, yet predicting AD progression in its initial phases will contribute to understanding this disease and preventing its progression (Li et al., 1999). An AD diagnosis requires various medical tests and a massive amount of diverse data. However, due to the heterogeneous nature of medical tests, manual comparison, visualization, and data analysis are difficult and time-consuming (Kimura et al., 2018; Blank, 2019).

Brain connectivity can be categorized as either functional, structural, or effective connectivity. Structural connectivity (SC) refers to the white matter tracts that physically connect two brain regions (Huang et al., 2009). The spatiality of brain regions does not necessarily define brain regions’ co-activation. Functional connections (FC) indicate the underlying co-activation of neural regions that are passively and positively connected and also identify the number of FC between brain regions (Friston, 2009). Thus, knowing how neurons and neural networks process information requires understanding brain connectivity. Effective connectivity quantifies the direction of an area’s influence on other functionally connected brain regions (i.e., node). It is crucial to discover the patterns of interaction between different brain regions, such as those in the DMN, and how AD affects these patterns of interaction, which reflect changes in the cognitive process (Kazemi-Harikandei et al., 2022; Zhong et al., 2014).

The article is organized as follows: ‘Materials and Methods’: Strategy and Methods used for the systematic review. ‘Background’: Background of Alzheimer’s disease. ‘AD Datasets’: Review of public datasets related to Alzheimer’s disease. ‘Preprocessing’: Standard preprocessing phases of fMRI data. ‘Functional connectivity (FC)’: Various kinds and analyses of functional connectivity. ‘Feature Extraction’: Methods for extracting features. ‘Feature Selection’: Techniques for selecting elements. ‘Machine learning (ML) models for classification of AD’ and ‘Deep learning (DL) models used to classify AD’: Machine learning (ML) and deep learning (DL) approach for detect Alzheimer’s disease in the literature. The article ends in the ‘Discussion’ and ‘Conclusion’.

Materials and Methods

Search strategy

Many recent research papers have highlighted the significance of artificial intelligence (AI) and its applications in the medical field for diagnosing various disorders, including Alzheimer’s disease (AD). In this review article, we utilized a PRISMA flowchart based on Page et al. (2021) to illustrate our study selection strategy, including the inclusion and exclusion criteria. A total of 534 articles were initially matched according to our search criteria (Fig. 1). We excluded the following papers: (1) duplicates, (2) unrelated to fMRI, (3) not written in English, and (4) survey papers. After these exclusions, we thoroughly examined the remaining 92 papers, with the top 81 candidate papers representing the literature related to AD classification.

Figure 1 Flowchart (PRISMA) depicting the process of searching for and selecting literature.

To gather relevant literature, we searched titles and abstracts of AD-related studies in the IEEE Xplore, Elsevier, and PubMed databases. Additionally, we used Google Scholar to ensure no publications were overlooked, using the search terms “FC in AD using fMRI data” and “Detect AD using AI” (on Jan 1, 2024). Figure 2 depicts the number of publications related to AD and their increase over the last five years.

Figure 2 The number of articles published in the different database using the search term FC in AD using fMRI data.

Research questions for systematic review

We aim to tackle the problems of the following research questions (RQ) after reviewing research in this field:

• RQ1: How does FC contribute to the detection of AD?

• RQ2: What are the neuro-functional interactions that might lead to early detection of AD?

• RQ3: How do ML and DL aid in detecting the early stages of AD?

Background

AD is a brain disorder that affects the memory of a patient. Doctors performed tests for the assessment of impairment and memory other than the skills of thinking for the diagnosis of AD (Wurtman, 2015). An AD diagnosis requires various medical tests and a massive amount of diverse data. However, due to the heterogeneous nature of medical tests, manual comparison, visualization, and data analysis are difficult and time-consuming (Kimura et al., 2018; Blank, 2019). Figure 3 shows the estimated AD costs for Medicare and Medicaid through 2050.

Memory loss, difficulties communicating with people, and repeatedly asking questions are some AD symptoms (Budson & Solomon, 2021). Importantly, we can classify AD into two categories: invasive procedures, which are of utmost importance in medical practice, involve entering the body, often with an instrument, to access internal structures directly. In the context of brain imaging, an invasive procedure might involve inserting a probe into the brain tissue for direct measurement or intervention; non-invasive procedures, on the other hand, do not penetrate the body or disrupt normal tissue. In the context of brain imaging, non-invasive procedures include techniques like MRI, CT scans, or EEG, where no physical penetration of the body occurs.

Figure 3 Estimated AD costs for Medicare and Medicaid through 2050 (Kimura et al., 2018).

Invasive approach

A brain autopsy is an invasive approach to diagnosing AD. AD has some pathological changes, such as Cerebrospinal Fluid Aβ Peptides. The hallmark of AD’s major pathology includes the β amyloid βpeptides extracellular deposition and metabolism, making it the most dominant neurodegenerative disorder. The abnormal aggregation of β-amyloid (Aβ) peptides, where solvable oligomers are converted into unsolvable fibers or plaques, is the primary mechanistic event, according to the hypothesis of amyloid cascade, which supports the pathophysiology of AD (Blennow et al., 2015). Cognitive decline and the onset of dementia are symptoms of persistent neurodegeneration brought on by the increasing Aβ plaque deposition, which damages neurons and impairs synaptic function (Alzheimer’s Association, 2018). The hypothesis of amyloid cascade, which holds that the formation and buildup of plaques is the main cause of AD, is supported by biochemical, pathological, and genetic data. The discrepancy between the creation and clearance of peptides of Aβ, such Aβ42, is thought to be the first step in the pathogenesis of AD, according to data from clinics and laboratories around the world (Richens, Lee & Johri, 2020).

Non-invasive approach

Non-invasive approach procedures can be used to measure the brain areas’ structures, development, and processes. This technique is safe and painless, and with its high resolution, it can be used to diagnose any disease in the brain. Image modalities differ in terms of image depth, resolution, cost, accessibility, and association with known features of disease activity. They are becoming increasingly important in assisting clinicians in diagnosis and guiding patient care and treatment, but this technique is susceptible to noise (i.e., artifacts) (Reiman & Jagust, 2012; Giakos et al., 1999). There are various neuroimaging scans to detect and classify brain pathologies (i.e., tumors, epilepsy, sleep disorder, and AD); see Fig. 4.

Figure 4 Types of neuroimaging in AD.

Magnetic resonance imaging

Magnetic resonance imaging (MRI) is a tomographic imaging technology based on the concepts of atomic nucleus resonance transitions between distinct magnetic energy levels. In this, absorption and re-emotion of electromagnetic radiation occur in atomic nuclei with a specified resonance frequency in an exterior magnetic field. This electromagnetic radiation is recorded to produce magnetic resonance signals. In medical applications, MRI employs the magnetic resonance signal of hydrogen atoms to view tissues or diseases in narrow slices through the human body (Bernstein, King & Zhou, 2004; Heidemann et al., 2003). The volume of the appropriate slice is shown in each image. Combined, these images produce a 3D depiction of the brain’s structure, including the volume, shape, and location of brain regions. (Yagis et al., 2020) employed a 3D variant of the VGG convolutional neural network (CNN) to investigate classification accuracy using the ADNI and OASIS datasets. To avoid information loss from slicing 3D MRI into 2D images, 3D models were utilized along with preprocessing techniques to enhance classification performance. Their model achieved 73.4% accuracy on ADNI and 69.9% on OASIS with five-fold cross-validation, outperforming 2D network models.

Functional magnetic resonance image

Functional magnetic resonance imaging (fMRI) is a popular method for measuring the primary visual cortex of the brain and for measuring brain topography. It examines the resting state of the brain, which means the patient cannot do any activity or perform a task-related paradigm in an indirect manner (i.e., stimulus, response) (Fleisher et al., 2009; Lajoie et al., 2017). The most frequent technique for measuring activity patterns is blood-oxygen-level-dependent (BOLD) imaging (Gusev et al., 2021; Ulmer & Jansen, 2010). BOLD fMRI is based on changes in blood oxygen levels (deoxyhemoglobin concentration) caused by changes in blood flow in brain sub-volumes called voxels, which is the 3D equivalent of a pixel. The spatial resolution for measuring brain activity is less than three seconds, and the spontaneous low-frequency fluctuation is 0.1 Hz in the BOLD signal (Ogawa et al., 1990).

sMRI and fMRI are essential in AD research, offering complementary insights. sMRI detects structural changes like hippocampal atrophy, providing detailed anatomy but no functional information (Burggren & Bookheimer, 2002), as shown in Table 1.

Table 1 Comparison between SMRI and fMRI.

	SMRI	fMRI	
Purpose	For examining the anatomy and pathology of the brain	Examine brain activity by measuring changes in blood flow.	
Performance	High spatial resolution	Lower spatial resolution, offers excellent temporal	
		resolution, making it suitable for studying dynamic	
		brain processes and functional connectivity.	
Role in AD detection	Used to detect structural changes associated	Reveals functional changes in brain activity patterns,	
	with AD, such as hippocampal atrophy,	including alterations in the default mode network (DMN)	
	a hallmark of the disease.	commonly observed in AD.	
Dimensional	3D (x, y, z)	4D (x, y, z, time)	
Types	T1-weighted MRI, T2-weighted MRI	Task-based fMRI (stimuli)	
		and resting-state fMRI (without stimuli)	

MRI and fMRI detect AD differently, especially in terms of how sensitive they are to early-stage changes. fMRI is more effective in detecting functional changes in the brain, such as blood flow and neural activity alterations, which can occur in the early stages of AD before significant structural changes are evident (Celone et al., 2006).

Sperling et al. (2009) demonstrated that fMRI could detect functional changes associated with AD in individuals with MCI, a condition that often precedes AD. This suggests that fMRI may be more sensitive to early-stage AD than structural MRI.

Positron emission tomography

Positron emission tomography (PET) is a functional scanning that produces a 3D image of the molecule and cell by radiotracer injecting and generating a digital image by utilizing a scanner (Reiman & Jagust, 2012). The evidence of clinical data suggests a relationship between glucose hypometabolism and amyloid deposition in AD, as revealed by fluorodeoxyglucose (FDG-PET) and amyloid PET, respectively (Kumar et al., 2022). This scan uses radiotracers to analyze the brain’s activities as radioactive spheres (Cohen & Klunk, 2014).

Computed tomography

Computed tomography (CT) scans utilize X-rays to detect brain shrinkage caused by dementia and to confirm the presence or absence of the disease (Cuttler et al., 2017). They are also helpful in diagnosing and monitoring other medical conditions, including cancer, heart disease, and neurological disorders (Scheltens, 1999).

Diffusion tensor imaging

Diffusion tensor imaging (DTI) is referred to as a famous technique for MRI (Alexander et al., 2007). The initial rules for MRI inferred in the 1980s when the researchers combined the principles of nuclear magnetic resonance (NMR) imaging with the diffusion effects of encoding molecular. The diffusion based on molecules refers to molecules’ random translational motion. This motion is also referred to as Brownian motion, which is the result of molecules’ thermal energy. This was implemented by using the gradient pulses of the bipolar magnetic field in the signals of NMR. DTI is utilized to visualize the structure of white matter (Assaf & Pasternak, 2008). The studies reflect the good potential of the method. The application is negatively affected by diffusion based on Gaussian.

Electroencephalogram

Researchers have widely used electroencephalograms (EEGs) for the past few decades to diagnose AD as it reflects a shift of the power spectrum to frequencies with lower values (Jeong, 2004). The EEG abnormalities also show the fast rhythm coherence. These abnormalities are also linked to functional disconnections among regions of cortical. These disconnections also lead patients to cholinergic deficits, axonal pathology, and cortical neuron death. The EEG is also utilized to assess the progression and diagnosis of AD (Tsolaki et al., 2014). The recordings of an EEG can also be used to add vital information to make the drugs more effective.

AD Datasets

AD Neuroimaging Initiative

Its open source dataset is retrieved from the AD Neuroimaging Initiative (ADNI). The ADNI started in 2004 under the leadership of Dr. Michael W. Weiner. The set was provided by the University of Southern California (USC). ADNI aims to develop biomarkers as clinical trial outcome measures. The ADNI includes MRI, fMRI, PET, DTI, and genetic data sessions at various stages for males and females (Alzheimer’s Disease Neuroimaging Initiative, 2023).

Open Access Series of Imaging Studies

Publicly available dataset retrieved Open Access Series of Imaging Studies (OASIS). The data from 1098 individuals were divided into 605 normal cognitive adults and 493 with different stages of declined cognitive adults and ages between 42 and 95 years. Data included MRI, fMRI, CT, and PET neuroimaging technique sessions collected from Washington University Knight AD Research Center over 15 years. It contains about 2000 MR sessions between SMRI and fMRI (Marcus et al., 2007).

Preprocessing

Due to the noise in fMRI data, image preparation is essential before feeding data to models or classifiers in all types of neuroimaging. Image preparation includes artifact removal, realignment, registration, slice time correction, spatial normalization, spatial smoothing, segmentation, and data augmentation to be prepared for model building to assist in obtaining accurate outcomes (Jenkinson & Smith, 2006; Smith, 2004), as shown in Fig. 5. To ultimately include recent advancements in MRI and fMRI preprocessing into a single coherent software package, (Park, Byeon & Park, 2019) suggested a new preprocessing pipeline, FuNP (Fusion of Neuroimaging Preprocessing) pipelines, which were applied to open and local datasets.

Figure 5 Standard preprocessing steps for fMRI data.

The first part of a pipeline of preprocessing techniques is to remove artifacts (Hutton et al., 2011). It is necessary to minimize data artifacts, mitigate noise effects, and compensate for any potential image degradation while collecting the data (Lindquist, 2008). According to empirical evidence (Skup, 2010), the signal of fMRI could be impacted via three main kinds of artifacts, including subject-related noise that is mainly caused by head motion, MRI hardware system noise, and physiological noise of fMRI imaging that probably causes signal loss, as shown in Fig. 6.

Figure 6 Signal before and after de-noising data.

Realignment

Image realignment aims to remove artifacts caused by the patient’s motion during the fMRI scan. This involves several linear transforms: shifting (translation), rotation, stretching (scaling), and shearing on all axes. The translation range should be between −2.5 and 2.5, and the rotation range should be between −1.5 and 1.5. Translation has three parameters x, y, and z, which are the head motion parameters at three different angles. After that, rotation in three parameters means moving the head in three different directions while keeping the head in the same position (Zafar et al., 2015).

Registration

In the registration step, various imaging modalities are needed (e.g., functional with T1 MRI). Further, in group analysis, it is supposed that each subject’s voxel is placed in the same location as other subjects (Zhao et al., 2017). Scanners have different parameters for scanning, such as slice count, slice thickness, flip angle, and so on. The fMRI scans must be registered on the image of reference using transformations based on geometry to make one image fit into another. Slice-by-slice matching ensures that all scans are similar to the idea of reference.

Slice time correction

Slice timing correction (STC) is a crucial step in fMRI data preprocessing, particularly for datasets with fast repetition times (TR). This correction accounts for the slight time differences in acquiring slices of BOLD data, which can impact the analysis (Parker & Razlighi, 2019). Slices may be acquired in ascending or descending order or using interleaved acquisition (e.g., odd or even slices) (Calhoun, Golay & Pearlson, 2000). Key parameters include slice order, the number of slices, echo time (TA), and TR that determines how frequently images are acquired over time, impact the temporal resolution of the scan. STC aims to synchronize the acquisition times of slices, enhancing the accuracy of the data for subsequent analysis. STC was calculated using Eqs. (1) and (2) (Parker & Razlighi, 2019). (1) TR=Time taken to acquire one volume of fMRI scan

(2) TA=TRTRNumberofslices.

Spatial normalization

Normalizing fMRI data is an essential step in analyzing the individual differences in brain activity patterns. Spatial normalization refers to aligning each subject’s brain to a specific point of reference. This step involves transforming each subject’s structural MRI scan (T1-weighted image) into a standardized template space, such as the Montreal Neurological Institute (MNI) or Talairach space. Various algorithms like linear affine transformations or non-linear deformations are used to achieve this alignment (Crinion et al., 2007).

Spatial smoothing

The spatial smoothing step (also called spatial filtering) is another preprocessing step to apply to fMRI data to increase the signal sensitivity and signal-to-noise ratio. To accomplish this, the full-width half maximum (FWHM) of a Gaussian kernel is used to compute a weighted average over multiple neighboring voxels at each voxel. The value is typically set to 4–6 mm for single-subject experiments and 6–8 mm for multi-subject analyses (Mikl et al., 2008).

Segmentation

The brain consists of a variety of tissues. It is essential to distinguish between white matter (WM), gray matter (GM), and cerebrospinal fluid (CSF) tissues when extracting signals of interest. According to the Gaussian mixture model, the chief objective when segmenting the brain fMRI is to divide the image into distinct regions. Each region should contain a set of pixels with the same range of intensities or texture (Sampath & Saradha, 2015).

• White matter (WM): refers to brain tissue composed of myelinated axons. It appears white due to the high myelin content. WM facilitates communication between brain regions and is important for cognitive function. Changes in WM integrity or connectivity are often associated with various neurological and psychiatric disorders (Oishi et al., 2010).

• Gray matter (GM): GM primarily comprises neuronal cell bodies, dendrites, glial cells, and capillaries. It plays a crucial role in processing information in the brain, including sensory perception, motor function, memory, emotions, and decision-making. GM appears gray in contrast to white matter due to its higher density of cell bodies and lower concentration of myelin, a fatty substance that surrounds and insulates axons in white matter. Myelin gives white matter its characteristic white color (Matsuda, 2016).

• Cerebrospinal fluid (CSF): CSF is a clear, colorless fluid that surrounds the brain and spinal cord. It acts as a cushion to protect these organs from injury, helps remove waste products from them, and maintains a stable environment for neural signaling. In fMRI, CSF appears dark on images and is typically used as a reference or baseline for analyzing brain structures and functions. CSF-filled spaces, such as ventricles and subarachnoid spaces, can be identified on fMRI scans and are essential landmarks for assessing brain anatomy and pathology (Blennow et al., 2010).

Data augmentation

It is possible to avoid overfitting training models by increasing data size and dealing with imbalanced data. Several operations can be generated from the original image in augmentation steps, such as rotation, translation, scaling, gamma correction, random noise addition, and random affine transformation (Garcea et al., 2022; Shorten & Khoshgoftaar, 2019).

These preprocessing pipelines are handled mainly by software tools like SPM (Friston, 2003), FSL (Jenkinson et al., 2012), CONN toolbox (Nieto-Castanon, 2020), and fMRIPrep (Esteban et al., 2019). The software package and the study’s objectives frequently determine the specific preprocessing techniques. Nevertheless, there are certain limits to the tools of fMRI preprocessing. In particular, devices for preprocessing can consume a significant time and computer resources. As a result, the number of subjects, hardware, and time available may dramatically lengthen the preprocessing stage. In turn, to save time, there is a need to use high-performance computers while dealing with the fMRI data.

Functional connectivity

Functional connectivity (FC) is used to measure temporal correlations among fluctuations of BOLD signal in various brain areas, represented in a matrix of size n × n for each member, where n refers to brain region numbers generated from atlas parcellation (Zamani, Sadr & Javadi, 2022; Van Den Heuvel & Pol, 2010). On rs-fMRI, the default mode network (DMN) is a brain network that has been frequently studied and is involved in tasks of memory consolidation (Mohan et al., 2016). It is made up of the posterior cingulate cortex (PCC), precuneus (Prec), lateral parietal cortex (LPC), retrosplenial cortex, inferior parietal cortex (IPC), medial parietal cortex (MPC), and medial prefrontal cortex (mPFC) (Chong, Schwedt & Hougaard, 2019). Here, we addressed RQ1.

AD patients have poor DMN connectivity (Grieder et al., 2018). There has been a steady indication of reduced FC in the DMN of the person suffering AD compared to HCs, particularly between Prec and PCC. For instance, the anterior cingulate cortex(ACC) and the mPFC (Brier et al., 2012; Gili et al., 2011; Griffanti et al., 2015). The discovered decrease in FC in DMN regions has also been stated in patients with MCI (Cha et al., 2013; Ouchi & Kikuchi, 2012).

Zhou et al. (2022), identify the abnormalities of structural connectivity(SC) and FC for DMN with fMRI and diffusion-weighted imaging (DWI) by using a dataset of 120 individuals. This method is also tested by using a replication data of 122 persons. The components of DMN are used to identify the disrupted FC and SC. Another study by Ahmadi, Fatemizadeh & Motie-Nasrabadi (2021) constructed undirected weighted graphs using AAL atlas-based fMRI data, and applied various kernels to compute the correlation of seeds from the AD and NC. Two statistical methods were utilized for the determination of the parameters. Global attributes of graphs are calculated to compare the kernel correlation analysis’s performance. The third-degree polynomial was considered to have the best version of all kernels. Jin et al. (2020) conducted FC and local activity in the DMN of AD and performed a meta-analysis on three groups of fMRI data. Additionally, Yu et al. (2019) concentrated on triple networks that contain the DMN, executive control network (ECN), and salience network (SN). Then, they applied Granger causality analysis (GCA) to detect casual impacts among the ROI of the brain to assess the three stages of AD. This proposes that rs-fMRI-detected variations in the DMN can be utilized as a non-invasive diagnostic tool for AD. Indeed, the National Institute on Aging-Association Alzheimer’s (NIAAA) has identified rs-fMRI. FC as probable bio-markers of neuronal damage that are still in the early validation stages (Albert et al., 2013).

The seed-based analysis (SBA) or region of interest (ROI) analysis is widely used by Jiang et al. (2004) and Jiang et al. (2014) due to its simplicity and straightforwardness, as well as the FC map clarity, which necessitates prior knowledge, making it challenging to examine FC correlations across entire brain regions. In Sun, Wang & He (2022), the AAL-90 atlas created an FC matrix representing a whole brain region. Also, the Pearson correlation was calculated between the two ROIs of the brain and the convolutional neural network (CNN) with residuals combined with multi-layer long short-term memory (LSTM) to classify various stages of the rs-fMRI data and got an accuracy of 93.5% for AD vs. NC.

Brain network access pattern analysis was used by Wang et al. (2018) to provide a reliable way for categorizing NC, MCI, and AD subjects in the context of size-restricted fMRI data samples. To create each subject feature vector (FV), the author chooses the DMN of ROIs were used to calculate the correlation coefficients among any potential ROI pairs. A regularized LDA method was used to lessen the noise effect brought on by the small sample size. The extracted features are then projected onto a 1D axis using the suggested regularized LDA. This process illustrates that optimization is a daunting task in the neuroimaging field.

Although SBA can locate brain regions that are functionally related to the first selected seed, it cannot fully describe the joint connections of various brain sections (Van Den Heuvel & Pol, 2010). Dimensionality reduction involves reducing the high-dimensional fMRI data to a lower-dimensional representation that captures the most essential information. This can be done by using techniques such as principal component analysis (PCA) or independent component analysis (ICA) to identify the most informative features (Viviani, Grön & Spitzer, 2005; Esposito et al., 2003).

Contrary to the SBA, the ICA has no predetermined seed region choice.The BOLD signal is separated into different time passages and associated spatial maps. The components that result in signals based on non-Gaussian and unrelated statistics to one another without using any prior knowledge or preconceived notions, ICA retrieves information of FC by finding the synchronous neuronal activity patterns between nodes (Fox & Raichle, 2007). According to Zhang et al. (2015), PCA aims to identify voxel-correlation regions (Shi et al., 2020), a new method for classifying AD patients based on FC. The analysis is based on fMRI data collected at activity voxels in the brain. Initially, ICA is utilized to detect activity voxels in healthy and unhealthy people’s data. After that, the computation of FCs is carried out in both groups. Then, the FCs With substantial variances are recognized by utilizing statistical analysis.

A similar effort by Buvaneswari & Gayathri (2021) included several analyses on the ADNI data set for the fMRI scan, such as a kernel-based PCA and support vector regression (SVR) that included t-distributed stochastic neighbor embedding (tSNE) and polynomial kernel-based tSNE. Furthermore, Kam et al. (2019) utilized a novel CNN framework for learning embedded features from brain functional networks (BFNs) while diagnosing brain diseases. Duc et al. (2020) generated spatial maps of functional 3D group ICA that can be utilized as regression and classification features. Furthermore, the Mini-Mental State Examination (MMSE) scores of AD patients in South Korea were evaluated using rs-fMRI data and regression algorithms in conjunction with a 3D-CNN deep learning architecture. The group-independent component analysis (ICA) was good at discriminating functional features of AD, and resulted in an accuracy 85.27%. Similarly, Qureshi et al. (2019), used ICA to extract FC features from rs-fMRI data and then applied 3D-CNN; this study obtained an accuracy of 92.30%. Buvaneswari & Gayathri (2023) proposed a novel method for classifying AD and MCI from rsfMRI using the ADNI dataset. Their approach involves preprocessing, feature extraction, PCA for dimensionality reduction, and kernel-based SVR for classification. By incorporating tSNE and polynomial kernel-based tSNE, the method effectively merges correlated features. The kernel-SVR method shows better accuracy 98.53% compared to existing models.

ICA is preferred over SBA because it can handle whole-brain FC analysis. However, the drawback of ICA is that it is frequently challenging to distinguish meaningful signals out of noise and changes in the individual components. Consequently, this makes employing ICA for between-group comparisons difficult.

The last way to analyze rs-fMRI, which is GTA, examines the entire network structure of the brain using precise spatial data. Firstly, the linkages among all active region pairs involving nodes or (N) and edges or (E) are determined by the spatial parcellation of the BOLD signal utilizing an entire brain topological mapping. A “node” is a specific region of the brain, whereas an edge shows direct and indirect relationships among two specified nodes (e.g., X,Y). A “hub” is also a node with an integrative function, reflecting the variety of a region’s cross-network FCs (Khazaee, Ebrahimzadeh & Babajani-Feremi, 2015). The following formula is used to determine GTA (Eq. (3); Gould, 2012): (3) GTA=N,E∀X,Y∈N

Lama & Kwon (2021) used GTA to diagnose AD at various stages, relying on the linear support vector machine (SVM) and the regularized extreme learning machine (RELM) for classification. The node2vec graph embedding approach is used to convert graph features to FV. (Zhang et al., 2020) FC on three scales, encompassing global metrics, nodal traits, and the modular properties of rs-fMRI images via GTA, and then applied a two-layer RF for classification. Zamani, Sadr & Javadi (2022) employed ANN algorithms to optimize data based on neuroimaging have many parameters. FC is measured using the rs-fMR dataset, which was investigated as a GTA. Mao et al. (2021) selected different subjects from MCI, AD, and NC subjects (matched for age and sex) and used rs-MRI scanning to evaluate the patients. The density of the brain’s short and long-range FC values was computed using the ultra-fast graph theory’s voxel-wise FC density (FCD) mapping method. The authors conducted voxel-based between-group analyses of FCD values to identify the cerebral regions with notable FCD changes. The authors analyzed Pearson’s correlation between abnormal FCD and several clinical factors. Lama, Kim & Kwon (2022) constructed brain networks from fMRI data via Pearson’s correlation. The brain network’s graph features were converted to FV using the Node2vec graph-embedding technique. The feature selection is applied among different methods such as Least Absolute Shrinkage and Selection Operator (LASSO), feature selection with Adaptive Structure Learning (FSASL), Local Learning and Clustering Feature Selection (LLCFS), and Pairwise Correlation-Based Feature Selection (CFS). In addition, ELM was applied to classify MCI, and AD, vs NC.

In addition, Khatri & Kwon (2022) explored the link of connectivity between several regions of brain using GTA to compute features of nodal (degree of nodal (ND), the path length of nodal (NL), and between centrality (BC)) as features of the graphic. Besides, the author extracted 3D patterns to compute the region’s coherence and used a t-test to assess a 3D mask that conserves voxels. The researcher applied SVM, and a comparison of SVM and Random Forest (RF) performances was performed. Penalba-Sánchez et al. (2023) evaluated the dynamic and static FC of rs-fMRI utilizing various methods and applied it to 116 ROIs for four participant groups. The authors extracted dynamic FC and FC Using Pearson’s correlation, sliding-windows correlation analysis (SWC), and point process analysis (PPA); GTA was also generated to investigate network segregation and integration. The data revealed a longer typical path length and a lesser degree of EMCI than the other groups. In contrast to HC and EMCI, LMCI and AD showed a higher FC in several places. A summary of some of the latest articles used analysis of FC in Fig. 7 and Table 2.

Figure 7 Summary of analysis FC for fMRI scan.

Table 2 Summary of FC analysis.

No.	Authors	Data source	FC analysis	Classifier	Target	Accuracy	
1	 Zhang et al. (2020)	rs-fMRI (ADNI)	SBA	AdaBoost	Multi-classification	75.86%	
2	 Kam et al. (2019)	rs-fMRI (ADNI)	ICA	3D-CNN	Binary Classification	76.07%	
3	 Qureshi et al. (2019)	rs-fMRI (ADNI)	ICA	3D-CNN	Binary Classification	92.30%	
4	 Shi et al. (2020)	rs-fMRI (ADNI)	ICA	linear SVM	Binary Classification	92.90%	
5	 Zhang et al. (2020)	rs-fMRI (ADNI)	GTA	Two-layer random forest	Binary Classification	92.90%	
6	 Duc et al. (2020)	rs-fMRI (ADNI)	Group ICA	SVM	Binary Classification	85.27%	
7	 Buvaneswari & Gayathri (2021)	rs-fMRI (ADNI)	PCA	SVRa	Binary Classification	98.53%	
8	 Lama & Kwon (2021)	fMRI (ADNI)	GTA	SVM	Binary Classification	98.91%	
9	 Zamani, Sadr & Javadi (2022)	fMRI (ADNI)	GTA	EA + ANN	Binary Classification	94.55%	
10	 Sun, Wang & He (2022)	rs-fMRI (ADNI)	SBA	CNN +LSTM	Binary Classification	93.5%	
11	 Lama, Kim & Kwon (2022)	fMRI (ADNI)	GTA	ELM	Binary Classification	96.95%	
12	 Khatri & Kwon (2022)	fMRI(ADNI)	GTA	SVM	Binary Classification	96.95%	
				RF		85.15%	
13	 Penalba-Sánchez et al. (2023)	rs-fMRI (ADNI)	SBA	–	Multi- Classification	–	
14	 Buvaneswari & Gayathri (2023)	rs-fMRI (ADNI)	ICA	Kernal-SVM	Binary Classification	98.53%	
Notes.

a Support vector regression.

Feature Extraction

There are four categories of features extracted from the neuroimaging modalities including subject-based, ROI-based, slice-based, and voxel-based approaches. Figure 8, and Table 3 shows how frequently these approaches are used in the selected literature. Table 4 shows the approaches of fMRI, each of which has its strengths and limitations.

Figure 8 A summary of various feature extraction in the literature.

Table 3 Summary of feature extraction with the fMRI.

No.	Authors	Computational techniques	Data source	Scan types	Model	Accuracy	
1	 Kazemi & Houghten (2018)	Slice	ADNI	fMRI	AlexNet	97.63%	
2	 Bi et al. (2018)	Slice	ADNI	fMRI	SVM	94.44%	
3	 Zhao et al. (2019)	ROI	ADNI	rs-fMRI	SVM	81.11%	
4	 Ramzan et al. (2020)	Slice	ADNI	fMRI	2D-CNN	97.92%	
5	 Shi et al. (2020)	Voxel	ADNI	fMRI	linear SVM	92.90%	
6	 Janghel & Rathore (2021)	Subject	ADNI	fMRI-PET	SVM, LD, DT, K-means	99.95%	
7	 Mousa, Zayed & Yassine (2022)	ROI	ADNI	rs-fMRI	SVM	98.2%	
8	 Khatri & Kwon (2022)	Voxel	ADNI	fMRI-sMRI	RF-SVM	96.95%	
9	 Kwak et al. (2022)	ROI	ADNI	fMRI	Dense-CNN	75.85%	
10	 Dai et al. (2022)	Voxel	ADNI	fMRI	Hybrid-CNN	95.3%	
11	 Chauhan & Choi (2023)	ROI	ADNI	fMRI	ELM	95%	

Table 4 Strengths and limitations of feature extraction.

Methods	Strengths	Limitations	
Sliced-based approach	- Minimizes networks’ complexity by avoiding dealing with a large amount of parameters over training.	- Spatial dependencies between adjacent slices are lost.	
ROI-based approach	- Easily explainable
- Features a low dimension
- Fewer characteristics can reflect can reflect the whole brain	- Has little information on the brain regions involved in AD	
Subject-based approach	- It is not necessary to identify the ROI
- More accurate results because all information in subjects is included	- Higher needs in terms of computational resources	
Voxel-based approach	- Get information of 3D brain scan	- It has a high feature dimensionality as well as a high computation load in dependently, it leaves the local information of modalities of the neuroimaging modalities.	

Slice-based approach

Slice-based approach is based on the supposition that some attributes of interest can be minimized to 2D images, thus minimizing the number of hyperparameters. A 3D brain scan was used in some studies to extract 2D image slices (Ebrahimighahnavieh, Luo & Chiong, 2020). While others focus on typical neuroimaging projections such as the sagittal, coronal, and axial plane.

Sarraf & Tofighi (2016a) and Sarraf & Tofighi (2016b) used a CNN to classify AD, where the test data accuracy reached 96.85%. To facilitate the detection of AD and identify various degrees of dementia, the researcher examined various DL models for the detection of AD using different data sets and employing transfer learning strategies. Ramzan et al. (2020) used several preprocessing steps to transform 3D rs-fMRI data into 2D images. The authors acquired 2D images for all fMRI scans, along with the height of the image and axis of time, and stored them as PNG files. Furthermore, Kazemi & Houghten (2018) used median axial slices from 43 axial slices of fMRI data. Bi et al. (2018) used a slice-based approach to extract features and a random SVM cluster to differentiate AD from NC in fMRI data.

Region of Interest based approach

The region of interest (ROI) based approach is employed to compute image features (i. e., activate) and classify brain elements such as the superior longitudinal fascicle, corpus callosum, and hippocampi (Poldrack, 2007). In this section, we answered RQ2.

In Alorf & Khan (2022), a method for multi-stage classification of AD is presented. The authors utilized the DL models on rs-fMRI dataset and extracted FC of the brain network. Different regions of high significance were identified using the autoencoder and CNN model, such as the lingual gyrus, precentral gyrus, supplementary motor, and frontal gyrus. Zhang et al. (2019) investigated the potential relationship between FC and brain perfusion. They also analyzed the problems of MCI, AD, and NC. In this study, the fMRI data is utilized to assess the brain perfusion. They observed that the FC of the medial frontal-cingulate and left inferior temporal gyrus was lower in AD compared to NCs. Billette et al. (2022) studied 499 participants with fMRI data. The authors analyzed the regions such as the hippocampus, entorhinal cortex, precuneus, and whole-brain voxels.

Novelty-related activity in the region of precuneus was shown to shadow a nonlinear design across the clinical spectrum of amplified AD risk. Chauhan & Choi (2023) examined the use of fMRI-based functional connectivity (FC) measures in combination with extreme learning machines (ELM) to classify AD stages. The study found that non-linear techniques such as maximal information coefficient (MIC) and extended MIC (eMIC) achieved high accuracies, with eMIC-based features performing the best. The results showed that the technique achieved 94% accuracy for distinguishing between CN and MCI, 81% for MCI and AD, and 95% for CN and AD. These findings demonstrate the potential of fMRI and machine learning techniques in improving AD diagnosis.

Pathology-based studies reflect that neurodegeneration in AD begins in the limbic system, temporal lobe, neocortical regions, and hippocampus (Braak & Braak, 1995; Mueller et al., 2005). The brain network access pattern analysis is adopted by Wang et al. (2018) to offer a reliable method for differentiating NC, MCI, and AD subjects in the context of size-restricted fMRI data samples. To create a feature vector for each subject, the researcher first chooses the ROIs from the network having DMN and computes the coefficients of correlation between any potential ROI pairs. Zhao et al. (2019) conducted static and dynamic FC, extracted GM and WM from rs-fMRI, then did a t-test to check the different feature subset methods and applied SVM to the classifiers AD vs. NC. Kwak et al. (2022) focused on the hippocampal region in the intermediate stage of AD, known as MCI in rs-fMRI data, and used DL to distinguish stable and progressive MCI stages with an accuracy of 75.85%.

A regularized LDA method to lessen the noise effect brought on by the small sample size. The extracted features are projected on a 1D axis using the suggested regularized LDA. Also, (Hu et al., 2016; Ju et al., 2017), 90 ROIs were obtained using fMRI data, and then the correlation between brain sections of each pair was computed to produce a correlation matrix. In addition to distinguishing different AD stages, the ROI, showing substantial fluctuations based on strength among extracted features of CorrTF various AD stages (Mousa, Zayed & Yassine, 2022). Additionally, Suk et al. (2016) chose 116 ROIs from fMRI images, utilized a deep model on each ROI mean intensities, and then looked for non-linear relationships between the ROIs in an unsupervised and hierarchical manner.

Subject-based approach

Some studies used a subject-based approach to extract features via fMRI data. For instance, Sarraf et al. (2017) used the CNN model to distinguish between AD vs. NC based on the specific age of rs-fMRI and MRI data from ADNI. Furthermore, Janghel & Rathore (2021) converted 3D-fMRI and PET images to 2D and resized them before using a DL to detect AD. Finally, several classifiers are employed for classification.

Voxel-based approach

The most direct methods are voxel-based, which use voxel intensity values from the whole 3D scan of the brain. The hippocampal subfield and volume of amygdala nuclei gained from SMRI were united with network features of the brain and parts of multi-measure attained from rs-fMRI to extract voxel features, which were then classified by different models (Khatri & Kwon, 2022). Besides that, most individuals with AD experience neuropsychiatric symptoms (NPS), previously linked to brain structure and function alterations. Raczek, Cercignani & Banerjee (2020) sought to determine whether NPS at the 3-year follow-up had any structural or functional brain correlations. The volume of the regional brain and resting state network activity were compared to the NPS as determined by the Neuropsychiatric Inventory (NPI). In Dai et al. (2022), a method for disease classification called brain-former is presented. The process is based on the general architecture of the transformer. The purpose of this research was to classify the fMRI using voxel-based details. The local cues and the voxel were used to build the BrainFormer by utilizing a 3D-CNN model. A single-stream model was used to aggregate the global and regional signals. The data from the multisite was used by utilizing a normalizing layer. Finally, a map based on the gradient visualization approach was used to locate the biomarker.

Feature Selection

A good feature selection technique is a critical component of an ML model when dealing with high-dimensional features. Thus, in this part, we illustrate various aspects of feature selection in the fMRI literature. Table 5 contains a summary of the various ways to select a feature. To enable feature selection, an Adaptive Neuro-Fuzzy Inference System (ANFIS) and Chaotic Binary Grey Wolf Optimization (CBGWO) were suggested (Anter et al., 2019) to identify MCI vs.NC of rs-fMRI and to minimize the number of features without sacrificing important classification information. On top of that, naive Bayes (NB) is a portion of the criterion with chaos theory. Zhang et al. (2021), suggested a new multiclass classification architecture for AD with insert feature optimization and fusion that depends on multimodal neuroimaging. Their architecture is built with three attitudes: (A) An l2, 1-norm regularization part collaborates with the hinge loss of multiclass. (B) To combine the supplementary data existing in each method, an lp-norm (1 < p < ∞) regularization part is used. (C) A theorem that converts the multiclass hinge loss minimization problem using the l2, 1-norm, and lp-norm regularizations to a former answerable problem of optimization and its confirmation is given. The author found that their method supports the global optimum after different experimentations and analyses.

Table 5 Summary of feature selection with the fMRI.

Authors	Scans types	Types of subjects	Feature selection	Accuracy	
 Zhang et al. (2021)	SMRI	AD vs. HC	multiple kernel learnin basedtype	97%	
	fMRI				
	PET				
 Anter et al. (2019)	rs-fMRI	AD vs. HC	Chebyshev CBGWO-ANFIS	86%	
 Jiao et al. (2019)	rs-fMRI	EMCI vs NC	SFN and DFN	91.13%	
 Hojjati, Ebrahimzadeh & Babajani-Feremi (2019)	SMRI	AD vs NC vs MCI	DCA	67%	
	rs-fMRI		SFC	56%	
 Sheng et al. (2021)	fMRI	AD vs NC	OCF or reverse order	95.5%	
 Sadiq, Yahya & Tang (2021)	rs-fMRI	AD vs NC	ALFF+fALFF ReliefF-mRMR	96.36%	
 Lei et al. (2021)	Rs-fMRI	LMCI vs. NC	LASSO	85%	
 Nguyen et al. (2019)	rs-fMRI	AD vs NC vs.CN	Hybrid MVPA-LASSO	96.70%	

Usually, researchers construct brain functional networks based on time series, disregarding the complex and dynamic interaction relationships between brain regions. So, features computed using this applicable network may be ineffective as disease biomarkers. To address this issue, Jiao et al. (2019) presented a multi-scale feature combination for early MCI that depends on global static features, moment features, and more refined features that may be flexibly extracted from dynamic, static, and high-order functional networks. Furthermore, SVM was used to discriminate between eMCI and NC.

In order to select essential features and classifiers from different stages of multicollinear fMRI data (Teipel et al., 2017), classical stepwise logistic regression and elastic net regularization were implemented. Consequently, regularized regression achieved a 0.70% accuracy, which is a good result compared to other models. Hojjati, Ebrahimzadeh & Babajani-Feremi (2019) introduced the GTA utilized to compute measures of integration and segregation of rs-fMRI and structural MRI (sMRI), then adopted two algorithms to select features and choose the most important of the features, which are the discriminant correlation analysis (DCA) and sequential feature collection (SFC). Lastly, SVM was implemented to categorize the subjects.

To enhance feature selection, Sheng et al. (2021) used the fMR dataset to detect the functional brain connectivity relationship among different AD stages. The author split functional and structural brain MRI data into 360 areas which were primarily located in the frontal lobe and insular cortex through features and various classifiers for binary and multi-class classification. SVM reached the highest accuracy when compared to decision trees (DT), KNN, and ensemble methods. Sadiq, Yahya & Tang (2021) utilized two ways to extract feature PCC, fractional ALFF (fALFF) and amplitude of low-frequency fluctuation (ALFF), then select important features via ReliefF and minimum redundancy and maximum relevance mRMR, SVM was used for discriminating on various stage of fMRI and achieved 96.36% accuracy. To construct the dFC in AD, Lei et al. (2021) applied the sliding window method. Local weight clustering was used to extract features, followed by LASSO to select essential features. SVM was utilized to differentiate between late and early MCI vs.AD and achieved 85% accuracy.

Nguyen et al. (2019) constructed FC and extracted 3D-nodes (i.e., regions) from two data sets (rs-fMRI and local data), then applied the support vector machine recursive feature elimination (SVM-RFE) with multivariate pattern analysis (MVPA) as well as most minor absolute shrinkage (LASSO), including uni-variate t-tests, to reduce the non-importance feature. Lastly, the researcher used three classifiers: ELM, linear SVM, and non-linear SVM, to distinguish between AD, MCI, and NC and compare them with each other.

Machine learning models for classification of AD

Machine learning (ML) is a sub-field of artificial intelligence that focuses on creating algorithms and models that allow computers to learn and make predictions or judgments without being explicitly programmed (Erickson et al., 2017). ML is becoming adept at detecting AD due to advancements in computer vision (CV) (Suzuki, Zhou & Wang, 2017). This section addressed RQ3 and describes how ML models are utilized to enhance the detection and prediction of AD (See Fig. 9).

Figure 9 The most widely used ML algorithms from 2018 to 2023.

For instance, SVM approaches are frequently used for inter-group categorization of AD based on the FC scores of the rs-fMRI, DTI, and SMRI datasets (Dyrba et al., 2015). The Gaussian process may be used to classify logistic regression on rs-fMRI data (Challis et al., 2015). Another approach by Lin et al. (2018) used resting-state FC (rs-FC) to predict the AD assessment scale through linear regression models and FC features using partial least squares regression. In addition, rs-fMRI imaging was subjected to GCA based on voxels to uncover differences in directed FC between the hippocampus and the rest of the regions of the brain (Xue et al., 2019). Similarly, Sheng et al. (2019), utilized connectivity information in rs-fMRI data and then applied multi-modal parcellation (MMP) to classify subjects at various stages of AD. Liu et al. (2020) highlighted feature extraction for cortical thickness and GM volume, shortest path length, and clustering coefficient using the AAL atlas for both rs-fMRI and MRI, as well as multi-task feature selection using the least absolute shrinkage and selection operator task-task relationship (MTFS-gLASSO-TTR) as feature selection, followed by multikernel SVM classification.

Yang et al. (2019) implemented the brain functions network to classify intermediate stages in AD biomarkers and used multiple time-points of rs-fMRI data by combining the fused sparse network (FSN) model with parameter-free centralized learning (PFC). The essential features selected by the similarity network fusion(SNF) method were then used to classify them using SVM. Shahparian, Yazdi & Khosravi (2021) presented a technique for AD detection using fMRI data. The work was conducted by computing the time series of particular anatomical areas using only a person’s fMRI and extracting relevant features using the latent low-rank representation approach. An SVM classifier was used to classify the samples into NC or AD, depending on the retrieved characteristics. The suggested approach had an accuracy rate of greater than 97.5%. Li et al. (2024) proposed a method based on brain structural and functional connectivity and SVM to distinguish between normal elderly individuals and those with MCI. The approach achieved relatively high accuracy 93.75% and sensitivity 93.75% for MCI classification. These results underscore the potential of brain structural and functional connectivity in providing crucial information for the assessment and management of patients with MCI.

Wang, Liu & Yu (2021) used preprocessing to remove noise from SMRI and fMRI data before constructing a component analysis network (PCANet) for FC analysis, extracting features with a 3D shuffle net, and combining them with kernel canonical correlation analysis (KCCA). SVM was eventually used to classify a variety of stages of AD, and the author got 91.9% for AD versus MCI in SMRI and 87.0% for fMRI. Amini et al. (2021) extracted features using PCA and robust multi-tasking then used various classifiers such as KNN, SVM, DT, linear discriminant analysis(LDA), and random forests (RF). Furthermore, the author used the CNN model to forecast the severity of AD and achieved 96.7% accuracy.

Deep learning models used to classify AD

Deep learning (DL) is an area of ML that focuses on creating and deploying multi-layered artificial neural networks. These neural networks are built to show the human brain’s structure and function, enabling them to learn and make decisions based on massive amounts of data (Shen, Wu & Suk, 2017). Early, precise detection of neurodegenerative brain ailments is key for successful patient care and disease progression. DL models have enhanced recent computer-assisted diagnosis (CAD) research based on neuroimaging data (Khojaste-Sarakhsi et al., 2022). This section aims to demonstrate the DL algorithms used in AD detection, as shown in Fig. 10.

Figure 10 Summary of various DL algorithms to detect AD from 2018 to 2023.

Clinical outcomes can only be improved by early detection of AD. The most promising results can be achieved using SMRI and rs-fMRI. Hu et al. (2022) used a method for DL based on fMRI and SMRI, which is a Granger causality estimator, to construct the connectivity of the brain. The researcher obtained an accuracy of 87.23% for SMRI and 78.72% for rs-fMRI. Another study by Wang & Lim (2022) used the zoom-in neural network (ZNN) model to compare patients with MCI caused by AD to those with NC. ZNN builds a feed-forward hierarchy out of a collection of zoom-in learning units (ZLUs) without back-propagation. The 90 neuroanatomical functional regions provided by the AAL-90 atlas were evaluated and found to be implicated in AD. The 140-time series rs-fMRI voxel values in one brain node (i.e., region) extract the ZNN’s characteristics. Han et al. (2024) applied a graph convolutional networks(GCNs) framework to classify MCI and predict dementia risk using resting-state fMRI data. The GCN outperformed baseline GCN and SVM, achieving an average accuracy of 80.3% and a highest accuracy of 91.2%. The model with individual FC slightly outperformed that with global FC.

Parmar et al. (2020b) developed a 3D-CNN based on rs-fMRI data to predict AD growth in one person. Patterns are extracted from neuroimaging data. None of the usual techniques were employed to extract features, and their trials show that a pretty simple DL model performs well in AD categorization. The researcher stated that with more training data, fMRI-based biomarkers might aid in the early diagnosis and categorization of AD; the accuracy for this study was 96.67%. Li, Lin & Chen (2020) proposed a 4D-DL system (C3d-LSTM) for AD classification that can concurrently utilize spatial and time-varying data by inputting 4D-fMRI data. First, in the fMRI image, the author used 3D-CNN to extract spatial data; the time-varying data was then combined with an LSTM network. The C3dLSTM model efficiently processed 4D fMRI data and collected the spatiotemporal aspect of fMRI scan for early AD detection; the accuracy for C3d-LSTM was 85.96%. Similarly, Luo et al. (2021), an applied CNN model with a multi-layer LSTM algorithm was used for MCI diagnosis and prediction in a local dataset. The Chinese AAL atlas was used to identify 116 regions and build an FC network matrix representing brain regions, which achieved 93.5% accuracy.

Parmar et al. (2020a) presented an AD detection approach to use the DL algorithms to test fMRI data for medical reaction mixture-containing applications. They demonstrates one such fMRI-DL aid in which the author applies a model to pressing down datasets to showcase the capturing and identification of AD using the enhanced 3D-CNN model. The authors extracted spatiotemporal properties to categorize patients to discriminate between MCI and AD in their early and late phases, to aid in the identification and diagnosis. The accuracy for this approach was 93%. Guo & Zhang (2020) proposed a unique autoencoder network to distinguish between progression disorder and natural aging. It is based on a biased neural network’s functionality and can detect AD. The system is evaluated using the rs-fMRI scan, providing 25% greater accuracy than others. Li, Song & Qian (2022) observed 90 brain areas for AD classification, and time-series characteristics were extracted from the brain’s regions through a feature weighted-LSTM network (FW-LSTM) on the rs-fMRI data. Then, a Pearson correlation was calculated among pairwise regions of the brain. The accuracy that the authors obtained was 78.81%.

Odusami et al. (2021) proposed a DL model for predicting different stages of AD using an fMR dataset with 138 cases on the binary classification detection scenarios. The author used ResNet-18 to train CNN models for feature extraction. and then used randomization to reduce the gap between extracted features. The proposed model obtained a good result compared with other models. Lu et al. (2022) created a system that classifies AD, MCI, and CN in fMRI data using FC throughout the brain rather than feature selection, then used ELM to classify. This framework is only suitable for a small amount of data. The highest accuracy that the researcher got was 96.85% for AD vs. CN. Bi et al. (2020b) presented two DL models in a convolutional learning method that learns depth node connectivity features, versus recurrent learning which relies on adjacent positional features. Notably, ELM enhanced the model, and the researcher got 95% accuracy. Likewise, Bi et al. (2020a) presented an aggregator ELM-based graph convolution of rs-fMRI data. Then, a graph neural network (GNEA) was designed to classify AD, NC, and MCI. Wang et al. (2022) implemented a dynamic hyper brain network in the ADNI and OASI datasts to classify AD and NC with a 99.09% accuracy; their results 3% result compared with other studies. Wang & Li (2024) introduced a dual-branch fusion model to analyze temporal sequences in the ADNI dataset. The model features a class activation sequence (CAS) branch to highlight temporal node functions and a time-domain local branch for local feature extraction. A fusion module integrates features at different levels to capture temporal contextual relationships. Their model outperformed other algorithms, achieving higher classification accuracy in various tasks, including early diagnosis.

Discussion

The hippocampus is a region in the brain’s temporal lobe that plays a crucial role in memory formation and retrieval. It is responsible for converting short-term memories into long-term memories and is involved in spatial navigation and learning. In AD, the hippocampus is one of the first brain regions to be affected by the disease. AD is a progressive neurodegenerative disorder characterized by the accumulation of abnormal protein deposits (amyloid plaques and tau tangles) in the brain, leading to cognitive decline and memory loss. The hippocampus is highly susceptible to these protein deposits, which disrupt its normal functioning. As a result, individuals with AD often experience significant memory impairment, particularly in forming new memories and recalling recent events.

Therefore, studying the hippocampus can provide insights into understanding how AD progresses and affects memory function. By examining changes in this specific brain region over time, researchers can better understand the underlying mechanisms involved in AD development and potentially develop more effective treatments or interventions targeting this area. In summary, due to its crucial role in memory processing and its vulnerability to Alzheimer’s pathology, monitoring changes in the hippocampus can be an essential tool for detecting and studying AD.

Table 2 shows that most studies that applied FC got higher accuracy when encountering a binary classification given AD vs. NC. This can be justified by the distinct separation between the two classes as opposed to when dealing with a classification task of AD vs. MCI. In fMRI studies, most used SVM to classify brain activity patterns, such as distinguishing between different AD stages. On the other hand, DL techniques such as 3D-CNN are becoming popular in fMRI data to extract relevant features and patterns from the 3D brain images before the classification task. So, technology is not necessary for extracting import features, It is clear that this and here DL was preferred over ML.

One study identified for the use of MVPA in AD (Nguyen et al., 2019) the lack of studies on this subject, including: fewer publications. Some possible reasons include:

1. Complexity and technical challenges: MVPA is a relatively new and complex technique that requires specialized knowledge and expertise in neuroimaging analysis. It involves analyzing patterns of brain activity across multiple voxels (3D pixels) rather than focusing on individual voxels. The technical challenges associated with MVPA, such as data preprocessing, feature selection, and classification algorithms, may limit its widespread adoption and application in Alzheimer’s research.

2. Limited data availability: Conducting MVPA studies requires access to large datasets with neuroimaging data from individuals with AD. However, acquiring such datasets can be challenging due to privacy concerns and limited funding for large-scale studies.

3. Focus on other techniques: Alzheimer’s research encompasses many neuroimaging methods, including EEG, PET scans, etc.

It is important to note that the field of neuroimaging research is constantly evolving. The number of publications on MVPA in AD may increase as the technique becomes more established and its potential benefits are further explored.

Conclusion

AD is an irreversible, progressive brain disorder that slowly damages memory and cognition, eventually causing loss of mental function. AD is one of the illnesses that cause brain damage. This systematic review outlines the challenges of detecting AD before and after neuroimaging. Followed by the presentation of the standard preprocessing phases of fMRI, describes FC, and explores how various brain regions connect with multiple analyses. Followed by a description of different ways to extract and select essential AD features with binary or multi-class classification and discussed public datasets that are popularly used in the AD literature. This investigation inspires more research toward fMRI in ML and DL models, which will ultimately help enhance the diagnosis of the potentiality of the AD occurrence for people on the AD spectrum. Given the complexity of the acquired fMRI images, it is noticeable that feature extraction (e.g., slice-based or regain of interest) and having a proper functional connectivity analysis contribute to the predictability of AD. Noticeably, many research articles utilized general feature extraction methods. In recent research works, a few functional connectivity approaches have been of interest to the fMRI research community. These fMRI-specific feature extraction and functional connectivity methods may yield better results and result in a more timely AD diagnosis.

Supplemental Information

Supplemental Information 1 PRISMA checklist.

Additional Information and Declarations

Competing Interests

Author Contributions

Data Availability

The authors declare there are no competing interests.

Maitha Alarjani conceived and designed the experiments, performed the experiments, analyzed the data, performed the computation work, prepared figures and/or tables, authored or reviewed drafts of the article, and approved the final draft.

Badar Almarri analyzed the data, authored or reviewed drafts of the article, and approved the final draft.

The following information was supplied regarding data availability:

The data is available at ADNI (https://adni.loni.usc.edu) and OASIS (https://www.oasis-brains.org).

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
