# Peer review of "fMRI-based Alzheimer’s disease detection via functional connectivity analysis: a systematic review"

_PeerJ Computer Science, doi:10.7717/peerj-cs.2302_

## Round 0.1 · original submission · Major Revisions

There are many corrections to be made, including serious grammatical corrections.

Reviewer 1 ·

Basic reporting

Not clear and ambiguous.
Major corrections in English are required.
Article structure is not satisfactory.
Introduction is not adequate.
Low-resolution images used.
Equations are not cited in text.

Experimental design

Study design is clear, but methods are not described properly.
Coverage is unbiased and follow PRISMA statement.
The organization of content need improvements.

Validity of the findings

Topic is timely and within the scope, but the conclusion is not adequate.
Conclusion should be rephrased to make it more clear.

Additional comments

1. Line 39: What is WHG (2 14)? Please clarify and mention in manuscript.
2. Line 40: Serious grammatical errors were observed. Some examples: Authors are talking about recent research, but in addition they mention that it “can be used”. Also “The healthy brain and the brain affected by AD can be depicted in Figure. 1” – should be written as “… is depicted in Figure. 1”. Correct grammatical errors throughout the manuscript.
3. Some sentences are too small. For example, “43 Also, we can detect AD via retina”; “202 Following that, STC can be applied to fMRI data.”, “481 Most algorithms used in ML”.
4. Look for definitions of all abbreviations used. For example, “SC are usually white gray matter”.
5. Frequent context switching is observed in lines 54-59. Author should rewrite them to relate sentences with the earlier one.
6. Line 60-65, the organization should either be presented in terms of running text or brief points. Do not mix different types of sentences.
7. The resolution of some figures (i.e., fig. 1, fig. 5, etc.) is poor. Replace it with a high DPI image.
8. Check open and close quotations carefully. For e.g., line 71, caption of fig. 3, etc.
9. Equations should be cited in text instead of mentioning “The following formula is used to determine slice time correction”.
10. Segmentation techniques should be mentioned in section 4.6
11. The caption of fig. 18 (Summary of analysis FC) looks incomplete.
12. Heading of section 8 should be corrected. Instead of “FOR CLASSIFICATION AD” it should be “FOR CLASSIFICATION OF AD”.
13. Lines 623-626 should be rephrased to make it clearer.
14. Conclusion is not encouraging. Should be rephrased.

·

Basic reporting

1. The title must convey its meaning appropriately, hence should be rephrased. As per my understanding, study is not about performing functional connectivity analysis of brain during fMRI. But it is about studying the techniques used to analyse already obtained fMRI images for detecting Alzheimer’s Disease.
2. Use of abbreviations is not recommended inside an abstract.
3. Poor language usage and lack of technical writing in the manuscript.
4. In Fig. 2 only measuring unit i.e “$” is used for the “Y axis”, also mention the scale used to measure data along the “Y axis” (Like in millions etc). Refer the original graph.
5. The use of first-person pronouns like “We” must be avoided.
6. Line No. 88-89: “Importantly, we can classify AD into two categories: invasive (i.e., brain scanning) and non-invasive (i.e., opening the head)”.
Reconsider the definition of invasive and non-invasive, they are associated with wrong meaning.
7. Use of confusing abbreviations must be avoided. Like for repetition time TR is used and for echo time TA was used in Line 202 and Line 207 respectively.
8. In Line 202 fast repletion time is abbreviated as TR but expression (1) defines it as “Time taken to acquire one volume”, clarification is required.
9. Expressions are marked but never cited in text. Like eq. 1 and eq. 2.
10. Include more researches performed recently, in your study.

Experimental design

1. The citations should be made in format (Author et.al, Year).
2. The reference to the original source of images used must be included and all such figures must be cited. Like in Fig.1 and Fig. 2.
3. In Line 210-211: “Normalizing fMRI data is an essential step in the analysis process to account for individual differences in brain anatomy”. But fMRI does not deals with brain’s anatomy, it deals with functional mapping of the brain. It is structural MRI that deals with the anatomy.
Statement needs reconsideration.
4. In Table. 1, add a column for data source. Do all these research use fMRI data only in compliance with your title? Further, it can be unfair to compare accuracies of various studies, if performed on different datasets with different features.
5. Structural MRI is also one of the most widely used neuroimaging techniques used to diagnose Alzheimer's disease. Try to draw some comparison between them, may be on the basis of performance or results obtained in different studies.

Validity of the findings

1. Conclusions are well stated and connected to original research questions.

---

## Round 0.2 · accepted · Accept

The paper was well improved and can be accepted.